# Multisystem Endothelial Inflammation: A Key Driver of Adverse Events Following mRNA-Containing COVID-19 Vaccines

**DOI:** 10.3390/vaccines13080855

**Published:** 2025-08-12

**Authors:** János Szebeni, Akos Koller

**Affiliations:** 1Nanomedicine Research and Education Center, Institute of Translational Medicine Semmelweis University, 1089 Budapest, Hungary; 2SeroScience LLC, 1125 Budapest, Hungary; 3Translational Nanobioscience Research Center, Sungkyunkwan University, Suwon 16419, Republic of Korea; 4Cerebrovascular and Neurocognitive Disorders Research Group, Department of Morphology & Physiology, Institute of Translational Medicine, HUN-REN SE, Semmelweis University, 1094 Budapest, Hungary; 5Research Center for Sport Physiology, Hungarian University of Sports Science, 1123 Budapest, Hungary; 6Department of Physiology, New York Medical College, Valhalla, NY 10595, USA

**Keywords:** mRNA vaccines, lipid nanoparticles (LNPs), COVID-19, comirnaty, spikevax, adverse events (AEs), post-vaccination syndrome (PVS), microcirculation, endothelial inflammation, endothelitis, immune response, innate immunity, adaptive immunity, spike protein, systemic transfection, vasculitis, autoimmunity, complement activation, inflammatory signaling, ionizable lipids, vaccine-induced pathology, functional mimicry, multisystem inflammatory response syndrome (MIS)

## Abstract

mRNA-LNP-based COVID-19 vaccines, namely Pfizer-BioNTech’s Comirnaty and Moderna’s Spikevax, were successfully deployed to help control the SARS-CoV-2 pandemic, and their updated formulations continue to be recommended, albeit only for high-risk populations. One widely discussed aspect of these vaccines is their uniquely broad spectrum and increased incidence of adverse events (AEs), collectively referred to as post-vaccination syndrome (PVS). Although the reported PVS rate is low, the high number of administered doses among healthy individuals has resulted in a substantial number of reported vaccine-related injuries. A prominent manifestation of PVS is multisystem inflammation, hypothesized to result from the systemic transfection of organ cells with genetic instructions for a toxin, the spike protein, delivered with lipid nanoparticles (LNPs). In this narrative review, we focus on endothelial cells in the microcirculatory networks of various organs as primary sites of transfection with mRNA-LNP and consequent PVS. We outline the anatomical variations in the microcirculation contributing to the individual variability of symptoms and examine the molecular and cellular responses to vaccine nanoparticle exposure at the endothelial cell level with a focus on the pathways of a sustained cascade of toxic and autoimmune processes. A deeper understanding of the mechanisms underlying mRNA-LNP-induced AEs and PVS at the organ and cellular levels is critical for improving the safety of future vaccines and other therapeutic applications of this groundbreaking technology.

## 1. Introduction

With over 5.3 billion doses administered globally, the mRNA-containing lipid nanoparticle (mRNA-LNP)-based COVID-19 vaccines Pfizer and BioNTech’s Comirnaty and Moderna’s Spikevax remain to be the recommended tool in high-risk populations for maintaining immunity against circulating SARS-CoV-2 variants. However, as the perceived threat of the virus continues to decline and concerns grow regarding the unusually broad spectrum and comparatively high incidence of adverse events (AEs) [1,2,3,4,5,6,7,8], several countries and regions have decided to discontinue the use of mRNA vaccines. In the United States, the Centers for Disease Control and Prevention (CDC) and the Food and Drug Administration (FDA) have revised their guidance: they no longer recommend COVID-19 vaccination for healthy children and pregnant women and have moved away from universal booster programs toward a more selective, evidence-based approach targeting high-risk populations [9,10,11,12,13,14].

Regarding the clinical and social implications of vaccine-induced AEs, the reported incidence rate in the 0.02–0.5% range [8,15,16,17,18] is considered as rare on an individual basis; however, the vast scale of global vaccination campaigns has resulted vaccine-related injuries in the millions. These adverse outcomes have been collectively described as post-vaccination syndrome (PVS) [1,2,3,4,5,6], a newly recognized condition in which individuals with persistent health issues or disabilities may qualify as an iatrogenic orphan disease [8]. In some cases, simultaneous inflammation of multiple organs can occur after mRNA vaccination, leading to a highly lethal condition known as multisystem inflammatory response syndrome [19,20].

Given the multicausal and multiorgan nature of vaccine-induced inflammation, numerous pathological processes are likely involved. This review, based on official academic databases, focuses on microcirculatory inflammation, especially on the role of the endothelium, as a central driver of inflammatory AEs. Better understanding of these processes is essential for improving the safety, and thereby the future success, of mRNA-LNP platform technology, not only for vaccination but also for other therapeutic purposes.

## 2. Spectrum of mRNA-LNP-Induced Inflammatory Complications

Table 1 shows a list of organ-specific inflammatory symptoms and illnesses that were reported in the publicly available 3-month safety surveillance report of Pfizer [15]. The advantage of using these data is that the list of AEs likely excludes symptoms caused by SARS-CoV-2 infection, as—in theory—the vaccine had not been administered to infected individuals. The list was compiled using data from 52 countries, minimizing the risk of regional bias. Artificial intelligence assisted in the process by scanning over 1200 entries for terms containing the suffix “itis”. Such list of 244 conditions, about 20% of all AEs, does not distinguish among acute or chronic inflammatory conditions due to bacterial or viral infections or autoimmune attack; it just shows that, essentially, all organs can be inflicted with different forms of inflammation (without indicating the severity and prevalence/incidence), starting sometimes within 3 months after vaccination.

The similar kinetics of mRNA vaccine-induced inflammatory AEs listed in Table 1 and numerous commonalties in these illnesses, such as the association with reactivation of certain viral strains (Table 2), point to a common, fundamental immune abnormality or combination of abnormalities underlying these conditions.

## 3. The Root Causes of mRNA-LNP-Induced Inflammatory Complications

A recent comprehensive theory addressing the mRNA vaccine’s toxicity conundrum attributed the phenomenon, at least in part, to plausible consequences arising from the inherent structural and functional properties of mRNA-LNPs [8,29]. The concerns related to the design of mRNA vaccines vaccine include (i) ribosomal synthesis of the antigen (spike protein), which fundamentally alters the natural process of antigen processing and presentation; (ii) extensive chemical modification of the mRNA, rendering the spike protein translation poorly controllable and mutation-prone upon codon–anticodon pairing; (iii) the use of a proinflammatory, fusogenic aminolipid in the LNP, which promotes widespread systemic distribution of the mRNA-LNP entailing transfection of non-target cells to produce a toxin; (iv) the chemical stabilization of the spike protein, despite its known multiorgan, polymodal toxicity; (v) the choice of LNP lipid composition with reduced nanoparticle stability in aqueous environments; (vi) the PEGylation of the LNP surface, despite PEG’s known immune reactivity and immunogenicity; and (vii) the recombinant synthesis of mRNA and the multi-step organic synthesis of vaccine lipids that may lead to trace contaminations, including functional DNA sequences. 

These vaccine features collectively predispose to collateral tissue damage caused by inflammation involving cellular and humoral autoimmune responses. Although nucleoside modifications (e.g., pseudouridine substitution) of the mRNA reduce recognition by innate immune sensors, such as TLR3, TLR7, TLR8, and retinoic acid-inducible gene I (RIG-I) [30,31,32,33,34], residual activation of innate immunity by the modified mRNA via RIG-I-like receptors remains a possibility under certain conditions [35]. More importantly, however, the proinflammatory nature of LNPs used to deliver the modified mRNA offsets the intended reduction in immune reactivity.

## 4. Components and Organization of Microcirculation Across Tissues

Microcirculation refers to the smallest blood vessels in the body (below ~200 micrometer), including arterioles, capillaries, and venules, which are collectively responsible for gas and nutrient exchange, waste removal, and immune surveillance at the tissue level. Some morphological features of cells in the microcirculation are illustrated in Figure 1.

## 5. Distinctive Microcirculatory Architectures Across Organs and Their Impacts on Vaccine-Induced Inflammations

Although all organs share a basic microvascular blueprint, the microcirculatory networks differ markedly due to a variety of anatomical, physiological, and functional factors. These include the density of the capillary network (Figure 1A), EC ultrastructure and architecture (Figure 1B,C), and endothelial subtype based on the tightness of intercellular junctions (Figure 1D) [37,38]. Additional control is exerted by local regulatory mechanisms involving the endothelium, smooth muscle and adjacent tissue cells, such as pericytes and parenchymal tissues (Figure 1A,C), or other resistance elements that modulate blood flow in response to tissue-specific functional demands [39]. These structural and functional variations render the microcirculation a highly specialized, tissue-specific interface between the bloodstream and parenchymal cells. Consequently, they critically influence the nature and extent of inflammatory responses. Accordingly, the localization and systemic spread of the spike protein is strongly determined by the special structures and function of microcirculation of various organs.

### Organ-Specific Microcirculatory Structures with Impact on Vaccine AEs

The differences in the development of AEs may be due to the differences in the microcirculatory networks of various organs as illustrated in Figure 2. To illustrate these differences with an example, the heart’s microcirculation is characterized by an exceptionally dense capillary network, with nearly one-to-one capillary-to-cardiac fiber ratio, which is higher than that in skeletal muscles [40,41]. This dense packing of capillary endothelial cells and cardiac muscle fibers is demonstrated in Figure 3, which presents a cross-sectional view of rabbit heart tissue [40].

Importantly, the ratio of capillaries to cardiac muscle fibers can increase to 2–3.5:1 as a result of physical training, which may help explain the susceptibility of overstressed athletes to sudden cardiac death in the presence of pre-existing cardiac conditions [40].

As for the coincidence of myocarditis and/or pericarditis, a condition referred to as myopericarditis [42,43,44,45,46,47], the high-density capillary network in the myocardium is tightly connected to the epicardium, the inner layer of fluid-containing pericardium [48,49]. This provides the structural basis for myopericarditis being one of the most frequent AEs of mRNA vaccines [8].

In the brain, the ECs in continuous capillaries are connected by tight junctions, forming the blood–brain barrier (BBB), which strictly regulates molecular and cellular trafficking (Figure 2B). This anatomical configuration is consistent with the emergence of localized inflammatory processes, underlying epileptic, cognitive, and other focal AEs, such as optic neuritis, Guillain–Barré syndrome, facial paresis, and more. [50]. The localized, restrained inflammation also helps explain the functional neurological disorders which are characterized by neurological symptoms without detectable abnormalities upon imaging with MRI or CT [50,51,52]. The situation is similar in the spinal cord, for example, in transverse myelitis, which entails localized motor, sensory, and autonomic dysfunction. Corresponding to these observations, a recent study investigated the expression of SARS-CoV-2 spike protein in the intima (endothelium) of cerebral arteries following Pfizer and Moderna vaccinations, and found that spike protein expression was detected in 43.8% of vaccinated patients, which persisted up to 17 months post-vaccination. Spike protein mRNA was derived both from vaccine and virus [53].

In peripheral nerves, the endoneurium, which encases the individual nerve fibers (axons) along with their myelin sheath, contains an extensive network of capillaries also with tightly bound ECs (Figure 2C). Here, too, the inflammatory signals spread locally, along the axons and the perineurial and epineural tissue, leading to demyelinating neuritis of different afferent or efferent nerves.

In the lung, microcirculation comprises an extensive and highly permeable capillary network that lies in close association with the alveolar epithelium (Figure 2D). This anatomical arrangement facilitates the transmission of inflammatory signals to both type I squamous epithelial cells and type II secretory cells, leading to widespread inflammatory involvement of the respiratory tissue. Consequently, typical manifestations of vaccine-induced inflammatory adverse events in the lung include alveolar infiltrates accompanied by respiratory distress, dyspnea, chest tightness, and hypoxia [54,55].

In the kidney, the microcirculation consists of two distinct capillary networks: glomerular capillaries, which form high-pressure filtration units, and peritubular capillaries, which surround the renal tubules and facilitate selective reabsorption and urine concentration within the nephron (Figure 2E). Inflammation of the ECs in both types of capillaries explain the nephrotic syndrome characteristic of vaccine injury, manifested in acute onset edema, hypoalbuminemia, and heavy proteinuria [56,57].

In the liver, the microcirculation is organized into sinusoidal capillaries with a discontinuous endothelium, allowing free distribution of inflammatory mediators among the blood and hepatocytes. Accordingly, the symptoms of vaccine-induced hepatitis reflect common hepatocyte dysfunction, manifested in acute cholestatic liver injury (biliary obstruction), abdominal pain, pruritus, fever, fatigue, anorexia, nausea and vomiting, laboratory deviations, and, occasionally, fatal outcome in patients with preexisting liver disease [58,59,60,61,62].

Given this diversity of AEs in different organ systems, the impact of endothelitis must be tissue-specific, likely responsive to targeted anti-inflammatory therapies.

## 6. The Journey of mRNA-LNPs from the Deltoid Muscle to the Sites of Inflammations

Intramuscular injection of mRNA vaccines is generally associated with immediate interaction with antigen-presenting cells (APCs) and other immune cells at the injection site and within the draining lymph nodes. These locations are considered the primary sites for T and B cell priming and have remained the central focus of most narratives concerning the immune mechanisms of mRNA vaccines [63,64,65,66]. However, it is well established that LNPs injected intramuscularly rapidly distribute throughout the body, reaching various non-immune organs and cell types, many of which can take up the nanoparticles.

Three main mechanisms have been proposed to explain how mRNA-LNPs reach the systemic circulation. One is an accidental, direct injection into a small vessel. Because the standard immunization protocol does not require aspiration or needle retraction prior to injection [67], inadvertent intravenous administration and fast entry of vaccine into the brachial vein may occur, particularly in regions with dense capillary networks, such as the deltoid muscle. Second, entry via the lymphatic system which may start through the blind-ended portions of lymphatic vessels, which are leaflet-like endothelial structures that open in response to increased interstitial pressure (Figure 1A). The 300 µL bolus of vaccine can transiently and locally raise tissue pressure above the typically low pressure within lymphatic vessels, promoting nanoparticle entry. Once inside the lymphatic network, rhythmic contractions and relaxations of the lymphatic vessel walls propel the nanoparticles through valve-separated chambers and lymph nodes of lymphatic vessels into the venous blood [68,69,70,71].

High-speed video lymphoscintigraphic measurements showed the velocity of lymph flow in the ~2–9 mm/s range, which implies that the small portion (about 3%) of Comirnaty nanoparticles that does not reside in the deltoid muscle and the draining lymph nodes reaches the blood within about 1 h [72,73]. As additional variables that contribute to the individual variation in vaccine pharmacokinetics, the pumping activity of lymph microvessels are delicately controlled by nitric oxide and prostaglandins released from the endothelium [68]. Figure 4 details the anatomical roadmap of mRNA-LNPs’ transport from the deltoid muscle to the blood.

A third potential route for LNP entry into the bloodstream involves concentration gradient-driven transcapillary transport from the interstitial space into capillaries, particularly in regions where vascular permeability is elevated due to the proinflammatory effects of the vaccine. This process can be considered a reversal of the enhanced permeation and retention (EPR) effect, which is exploited for passive targeting of liposomes in cancer chemotherapy [74]. As for the extent and kinetics of systemic biodistribution of mRNA-LNPs, Pfizer Australia’s preclinical study reported that 2.8% of a radioactive lipid marker remained in the plasma of rats 15 min after intramuscular injection of Comirnaty-equivalent LNPs, with plasma levels peaking between 1 and 4 h [75].

## 7. Stoichiometry of Vaccine-Endothelium Interactions

The vascular endothelium, covered with glycocalyx, constitutes the first biological barrier to the systemic distribution of mRNA-LNPs, prompting the key question of stoichiometry: how many mRNA-LNPs actually reach and interact with the luminal surface of endothelial cells? A calculation at 15 min post-injection suggests that approximately 500 to 7000 mRNA-LNP may potentially interact with each endothelial cell in the body [75,76,77,78,79,80,81,82]. This implies that a single vaccine dose delivers mRNA-LNPs in substantial excess relative to the number of endothelial cells. The assumptions underlying this estimate are as follows: (i) each 30 µg dose of mRNA in a Comirnaty injection contains approximately 2 × 10^−8^ moles of mRNA, based on a molecular weight of ~1.4 MDa. This corresponds to roughly 1.2 × 10^13^ mRNA molecules; (ii) approximately 3% of the injected mRNA-LNPs reach the bloodstream within 15 min [75]; (iii) each LNP carries 1–2 mRNA molecules [76,77,78,79,80]; (iv) the total endothelial surface area in the human body is 1–7 m^2^, equivalent to 1–7 × 10^12^ µm^2^; and (v) the apical surface area of a typical endothelial cell is 300–600 µm^2^ [81,82].

It must be emphasized, however, that this is a snapshot calculation; the number of mRNA-LNPs available for interaction with endothelial cells rapidly declines with time due to disintegration and uptake by various organs, and it also depends critically on the rate of entry into the circulation. The key implication of these metrics is that vaccine-induced endothelial transfection, activation, and consequent inflammation could potentially affect any segment of the vascular endothelium. Hence, the small volume of vaccine inoculum does not preclude the possibility of off-target distribution of vaccine nanoparticles.

## 8. The Pathophysiology, Diagnosis and Example of Vaccine-Induced Vasculitis

### 8.1. Pathophysiology

Table 3 shows the types and symptoms of vasculitis classified according to the size of the blood vessels affected. Based on the proportional relationship between LNP uptake and capillary surface area, most vasculitis symptoms are associated with inflammation of small vessels, which collectively provide the largest endothelial interface. Nevertheless, the inflammatory diseases involving tissue compartments enriched in medium- and large-sized blood vessels have also been documented to display distinct clinical features [83,84,85,86] (Table 3).

### 8.2. Diagnosis

The diagnosis of vasculitis-related illnesses at the organ level remains challenging. Nonspecific symptoms such as fever, fatigue, and rash may not identify the affected organ, nor do inflammatory markers (e.g., CRP) and autoimmune indicators (e.g., ANCA, ANA) in blood or cerebrospinal fluid reliably pinpoint organ involvement. Imaging studies, including conventional or digital subtraction angiography, MRI with contrast and tissue biopsies support the diagnosis, but a thorough differential diagnosis is essential to exclude infections, malignancies, thrombotic disorders, and ischemic or hemorrhagic events [87]. Advanced techniques like vessel wall imaging (VWI) and magnetic resonance angiography (MRA) may reveal vessel abnormalities [88].

### 8.3. Experimental Evidence of Spike Protein-Induced Cerebral Abnormality

Focusing on the role of the spike protein in post-vaccination cerebral arteritis, a recent study demonstrated its prolonged presence in cerebral arteries, accompanied by inflammatory cell infiltration [53]. This chronic cerebral inflammation may disrupt local circulation and oxygen delivery, potentially underlying symptoms such as fatigue, memory loss, dementia, Alzheimer acceleration, and other cognitive or psychological complaints following vaccination [39,89]. Supporting this, we observed impaired cerebrovascular regulation in individuals with post-COVID condition (whether from infection or vaccination), assessed using transcranial Doppler ultrasound, a non-invasive method that measures blood flow velocity in major cerebral arteries. As shown in Figure 5, in individuals with persistent cognitive and mental dysfunction, even ~2 years after mRNA vaccination, it displayed diminished vasodilatory capacity of cerebral resistance vessels in the brain.

## 9. Non-Target Organ Uptake of mRNA-LNPs

### 9.1. The Role of Ionizable Lipid in Multiorgan Transfection

A key factor in selecting the ionizable, positively charged aminolipid ALC-0315 as the main lipid component of Comirnaty was the earlier clinical success of Onpattro (patisiran), the first FDA-approved gene therapy for amyloidosis, which also utilizes an ionizable lipid, DLin-MC3-DMA (MC3), to deliver a small inhibitory RNA (siRNA) into the liver [90,91,92,93,94,95]. It has been known since the late 1980s that such lipids tightly bind to negatively charged nucleic acids and deliver them into various cell types without loss of function [78,90,91,92,93,94,95,96,97,98,99,100,101,102,103,104,105,106,107,108]. Such transfection ability of LNPs has been linked, among others, to the fusogenicity of ionizable lipids at low pH, enabling endosomal escape of the nucleic acid into the cytoplasm [78,90,91,92,93,94,95,96,97,98,99,100,101,102,103,104,105,106,107,108]. Thus, the essential-and perhaps irreplaceable- role of ionizable lipids in therapeutic lipoplexes lies in their ability to complex with nucleic acids and facilitate their passage through bilayer barriers, particularly the endosomal membrane. As for their role in the cellular uptake of Comirnaty and Spikevax at neutral pH, it remains to be clarified to what extent ionizable lipids contribute via modulation of particle morphology, lipid packing, surface hydrophobicity, serum protein adsorption (i.e., formation of the protein corona), or recognition by scavenger and ApoE receptors.

### 9.2. Systemic Transfection vs. in Loco Immunogenicity of mRNA-LNP-Derived Spike Protein: The Achilles’ Heel of mRNA Vaccines

Gene delivery using LNPs has been observed in a broad range of tissues and cell types [98]; therefore, the assumption that Comirnaty delivers mRNA exclusively to phagocytic antigen-presenting cells (APCs) in the deltoid muscle and draining lymph nodes might have overlooked the possibility of mRNA dissemination. Among prior information, a 2015 study by Pardi et al. showed that the LNP-delivered firefly luciferase mRNA was expressed not only at the injection site and in lymph nodes but also in the liver, lungs, and other organs within five hours after intramuscular injection [99]. Pfizer Australia’s preclinical rat study similarly documented the distribution of LNPs to the liver, adrenal glands, spleen, and ovaries within 48 h post-injection, with low-level (<2%) radioactive signals detected in 12 additional organs [75]. In a porcine study, Ferraresso et al. [98] demonstrated that intramuscularly injected LNPs transfected nearly all major organs, while in another porcine study, Dézsi et al. reported both spike protein mRNA and the translated protein in multiple organs—including the heart and the brain—within six hours following intravenous injection of Comirnaty [100]. The efficient delivery of functional mRNA by LNPs into various organ cells has been well documented in many more studies [101,102,103,104,105,106,107,108].

### 9.3. mRNA-LNP Internalization by Endothelial Cells

As for the mechanism of mRNA-LNP internalization by non-phagocytic cells, such as the ECs, uptake of the nanoparticle may occur via multiple pathways, including clathrin- or caveolae-mediated endocytosis or micropinocytosis, especially in inflamed or activated endothelium. Once internalized, the mRNA and the newly synthesized spike protein can activate ECs through multiple, possibly synergistic signal transduction cascades. In this context, a key finding of the study by Dézsi et al. [100] is that mRNA uptake closely coincides with the transcriptional upregulation of major proinflammatory cytokine genes, including IL-6, TNF-α, and IL-1. This suggests intracellular crosstalk between the spike mRNA and innate immune signaling pathways, consistent with the combined proinflammatory effects of the mRNA, the ionizable lipid component of the LNPs, and the accumulation of spike protein in the cytoplasm and plasma. However, the relative contribution of each of these factors to organ-specific inflammation remains to be determined.

## 10. Endothelial Cells in the Frontline of Systemic Inflammation

Endothelial cells, being an interface between blood and tissues, are particularly responsive to inflammatory stimuli and are often the first point of contact for systemically circulating immune mediators and nanoparticles, including mRNA-LNPs. The causes of inflammation and the manifestations of symptoms vary widely across organs, reflecting not only differences in immune responses but also organ-specific anatomical factors, as discussed earlier in Section 5. The factors contributing to and consequences of multisystem endothelitis are illustrated in Figure 6 and detailed in Table 4.

The schematic diagram in Figure 6 outlines the major contributing factors and processes involved in endothelitis, which are detailed in Table 4 in terms of type, mechanism and consequences.

### 10.1. Spike Protein Toxicity on Endothelial Cells

The blood-borne, circulating spike protein, free or in exosomes [146], functions as a pluripotent toxin capable of inducing a range of deleterious effects in susceptible cells. These include oxidative mitochondrial damage [129,131,132], upregulation of proinflammatory cytokines [120,121], and other self-destructive processes in the cells that produce it [112,116,117,118,119].

Focusing on EC as a target for extracellular spike protein toxicity, recent in vitro studies using cultured human ECs demonstrated that the S1 subunit increases the production of inflammatory cytokines and induces NF-κB activation via binding to ACE2 and/or C3a receptors [116,119,147]. Additionally, the S1 subunit promotes microparticle (exosome) formation, a functional marker of endothelial injury [147]. These findings clearly support the capacity of circulating spike protein to induce endothelitis.

### 10.2. Vaccine-Induced Autoimmune Damage on Endothelial Cells

#### 10.2.1. Cross-Presentation Behind Cytotoxic T Cell Attack

Spike protein toxemia is only one risk for endothelial damage after mRNA vaccines. Another major source of damage is autoimmune attack against these cells, an ultimate consequence of unnatural intracellular processing and presentation of the spike protein. Although the spike protein is technically an exogenous (external) antigen, due to the fact that it is synthesized endogenously on ribosomes, it behaves like internal antigens. Specifically, the spike protein processing follows the endogenous antigen route, involving the degradation of the protein in proteasomes into short peptide fragments (~8–10 amino acids), transportation into the endoplasmic reticulum (ER) via the TAP (transporter associated with antigen processing) complex. Inside the ER, these peptides are loaded onto MHC Class I molecules with the assistance of the peptide-loading complex, and the resulting MHC I-peptide complexes are transported to the cell surface via the Golgi apparatus for recognition by CD8^+^ cytotoxic T lymphocytes (Tc-s). Because under physiological conditions, exogenous antigens are presented on MHC Class II molecules to CD4^+^ T cells, the above mistargeted presentation of spike peptides on MHC Class I proteins implies cross-presentation [148]. In individuals previously infected with SARS-CoV-2 or vaccinated with mRNA vaccines, spike-specific cytotoxic T cells may recognize and attack the ECs displaying spike-derived peptides on MHC I molecules (Figure 7), a mechanism likely contributing to the widespread autoimmune phenomena reported after mRNA-type vaccination, including the autoimmune damage of EC. The above spike protein-specific Tc-induced attack is consistent the increased incidence and severity of AEs following booster(s) compared to primary immunizations.

The endoplasmic reticulum (ER)-bound ribosomes, which give this large, membrane-bound organelle its “rough” appearance and represent a minority compared to free ribosomes in most cells, produce proteins destined for insertion into the plasma membrane or for secretion. Accordingly, in vaccine-transfected ECs, a portion of the spike protein becomes incorporated into the plasma membrane, mimicking true viral infection. The “spike-crowned” luminal surface of these “pseudo-infected” ECs elicits immune recognition, particularly by spike-specific antibodies generated from prior infection or vaccination. Antibody binding to the cell surface then initiates immune effector mechanisms, including complement-mediated (Figure 8A) and antibody-dependent cellular cytotoxicity (ADCC) (Figure 8B).

#### 10.2.2. Complement-Mediated Cytotoxicity

Complement-mediated cytotoxicity is executed by the membrane attack complex (MAC), formed by complement components C5b, C6, C7, C8, and multiple C9 molecules, which assemble into a pore-like structure that inserts into the target cell, in this case, the EC membrane. This disrupts endothelial membrane integrity, causing pathological ion flux, osmotic imbalance, and ultimately cell lysis or activation, depending on cell type and context [149,150]. In ECs, the MAC can induce sublytic activation rather than full lysis, leading to proinflammatory and prothrombotic responses such as increased expression of adhesion molecules, cytokine release, and tissue factor expression. These effects contribute to endothelial dysfunction, a key factor in vascular inflammation and thrombosis [151,152] as shown in Figure 5 and Table 4.

Complement activation by surface-exposed spike protein can be initiated via all 3 activation pathways (Figure 8A(a–c)). These are the classical, via the binding of anti-spike protein antibodies (Figure 8A(a)), the alternative, via direct binding of C3 to the injured membrane [110], and the lectin pathway, due to the mannose-rich glycosylation of the spike protein [111,112] (Figure 8A(a–c)). All these activations can be perpetuated via the alternative pathway amplification loop [153,154] fed by activated EC-produced C3, properdin and factor B.

It should be noted regarding complement activation by mRNA vaccines that despite early recognition [80,155] and increasing evidence of its essential role in inflammatory AEs, particularly anaphylaxis [156,157,158,159,160,161], this omnipresent, ancient arm of the immune system has been largely overlooked in the mainstream literature on vaccine-induced inflammatory AEs.

#### 10.2.3. Antibody-Dependent Cellular Cytotoxicity (ADCC)

The other antibody-mediated autoimmune EC damage is called antibody-dependent cellular cytotoxicity (ADCC) (Figure 6), whereupon effector cells of the immune system recognize and eliminate target cells that have been flagged by specific antibodies. The cells involved in this action include natural killer (NK) cells, macrophages, and neutrophils. The specific antibodies, usually IgG1 or IgG3, bind to antigens expressed on the surface of a target cel,l and these antibodies are recognized by the Fcγ receptors (especially FcγRIIIa/CD16) on the above cells (Figure 8B). Binding of FcγR triggers degranulation of these effector cells, just as the binding of CTL to MHC Class I molecules triggering the release of cytolytic mediators, namely perforins and granzymes. The latter proteins, especially granzyme B, are serine proteases that activate caspases (e.g., caspase-3) that cause DNA fragmentation and apoptosis. Granzyme A induces caspase-independent cell death (via reactive oxygen species, mitochondrial damage). Both NK cells and CTLs use receptor-ligand interactions leading to cell death, such as the binding of Fas Ligand (FasL, CD95L) to Fas receptors (CD95) on target cells, activating caspase-8 and causing apoptosis [162,163]. The TNF-a-related apoptosis-inducing ligand (TRAIL) is also involved in the apoptotic attack of Tc and/or NK on the ECs [164]. Among the cytokines produced, IFN-γ enhances macrophage activity and antigen presentation [165].

#### 10.2.4. LNP-Induced Inflammatory Signaling in Endothelial Cells

On top of the above mentioned cellular and humoral damages exerted on transfected endothelial cells, the known proinflammatory effect of LNPs also inflicts these cells activating diverse components of innate immune signaling, such as the NLRP3 inflammasome, Toll-like receptor TLR2 and TLR4, NF-κB, IRF3/7, MyD88, with downstream secretion of inflammatory cytokines, chemokines and adhesion molecules [166,167,168]. Products of mRNA-LNP-induced complement activation, C5a, C5b-9, are also known to interact with endothelial cells, triggering among others, NF-κB signaling [156,158].

## 11. Outlook

### 11.1. Vaccine-Induced Pseudo-Infection: Functional Mimicry of Pathogenicity in a Few?

Since the introduction and widespread deployment of mRNA-based COVID-19 vaccines, their safety profile has become a focal point of both public attention and scientific investigation. This cutting-edge technology has raised several unresolved questions, particularly regarding the unusually broad spectrum and relatively high incidence of AEs associated with their use [8,29]. According to an evolving paradigm, beyond their intended antiviral effects, COVID-19 mRNA vaccines may induce collateral tissue damage due to fundamental design limitations -primarily by altering the natural pathways of immunogenicity [8,29]. As a result, the unique properties of the spike protein antigen and its carrier LNP can elicit immune responses that resemble those triggered by active SARS-CoV-2 infection. Functionally, the vaccine nanoparticles act as simplified mimics of live virus, sharing some characteristics with live attenuated virus-based vaccines. The reversion to virulence is a phenomenon well documented for such vaccines, e.g., oral polio and yellow fever 17D vaccines [169,170]), but it is unambiguously excluded in the case of mRNA vaccines, as they do not contain replicating organisms capable of mutating back into a virulent form [171,172]. However, these vaccines do produce and disseminate a key pathogenic component of SARS-CoV-2, the spike protein, suggesting a form of functional pathogenicity. This hypothesis is supported by several observations: (i) the similarity between vaccine-induced adverse effects and COVID-19 symptoms, rationalizing the distinction of “Adverse Events of Special Interest” (AESIs) among the AEs of mRNA vaccines by the Brighton Collaboration [17]; (ii) the persistence of vaccine mRNA and protein for weeks-to-months post-injection [53,146,173,174,175,176]; and (iii) the theoretical possibility of reverse transcription of mRNA into DNA by human DNA polymerase theta (Polθ, EC 2.7.7.7) [176]. It needs to be emphasized, however, that this remains a hypothesis which challenges the prevailing scientific consensus regarding the mechanism of action of mRNA vaccines.

### 11.2. Scope and Justification

Despite the accumulating evidence of mRNA vaccine-associated AEs and their extensive scientific analysis and passionate public discourse, the biological basis of these complications remains poorly understood. This review aims to address that knowledge gap by analyzing the root causes and possible contributors to the AEs, with particular emphasis on microcirculatory endothelial inflammation as a potential key driver of multiorgan injury. While such focus may give the impression of downplaying the benefits of mRNA vaccines in halting the SARS-CoV-2 pandemics, and the broader promise of the underlying technology, this is not the authors’ intention. The emphasis solely on safety concerns reflects the need to stay focused on the defined objective of the review. We believe that investigating the mechanisms of vaccine-related AEs will contribute to the development of safer and more effective mRNA-based vaccines and therapeutics. A key unanswered question for future research is what protective factors prevent most vaccine recipients from adverse effects and post-vaccination syndrome.

### 11.3. Limitations of the Review

The review advances a mechanistic link between mRNA-LNP vaccines and microvascular inflammation but is limited by its reliance on hypotheses, preclinical findings, absence of large-scale epidemiological data, and limited consideration of alternative interpretations. However, several of these neglected aspects have already been explored, at least partially, in prior reviews [2,7,8,25,29,47,50,85,142,161], so their more detailed discussion was beyond the scope of the present broad, narrative overview.

## 12. Conclusions

In this review, we advance the ideas that the large surface area of the microvascular endothelium is the primary site of interaction with mRNA vaccine nanoparticles and that the resulting endothelitis and vasculitis may underlie the various symptoms of post-vaccination syndrome in a small subset of vaccine recipients. We outline several potential mechanisms whose root causes can be traced to unique properties of the mRNA-LNPs. A better understanding of these adverse reactions is crucial for their prevention and for improving the safety of future vaccines and other therapeutic applications based on mRNA-LNP technology.

## Figures and Tables

**Figure 1 vaccines-13-00855-f001:**
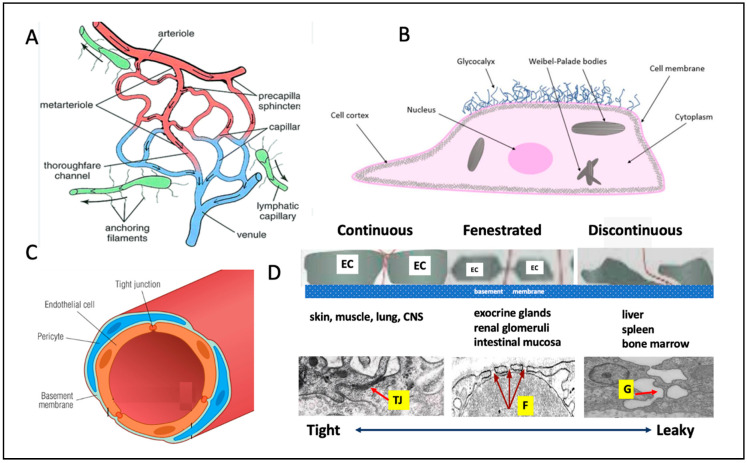
Schematic representation of microcirculation. (**A**) The arterioles and venules are connected by a network of capillaries. Some of the precapillary arterioles and capillaries have pericytes, specialized contractile cells (sometimes referred to as precapillary sphincters) that control the diameter of microvessels, and thus blood flow through the microcirculatory network [36]. In the interstitial space dead-end lymphatic capillaries are anchored to the underlying tissue by filaments. (**B**) Endothelial cells are composed of a plasma membrane with surface receptors, a cytoskeleton that maintains shape and enables signal transduction, cell junctions for barrier function, and organelles like the nucleus, mitochondria, and endoplasmic reticulum to support metabolism, protein synthesis, and regulatory functions. The glycocalyx is a thin, gel-like layer of glycoproteins and glycolipids that coats the surface of ECs, playing a crucial role in vascular permeability, mechanotransduction, and protection against shear stress and inflammation. (**C**) Cross-section of a capillary, illustrating the spatial arrangement of structural components such as the ECs, basal lamina, pericytes, and tight junction. (**D**) The three types of capillaries specialized for different organ systems; schematic illustration (top); typical electron microscopic images (bottom). From left to right: continuous capillaries with tight junctions (TJ), fenestrated ones with pores (F), and discontinuous capillaries with large gaps (G) between the cells. The illustrations are modifications of publicly available sources, including Google images.

**Figure 2 vaccines-13-00855-f002:**
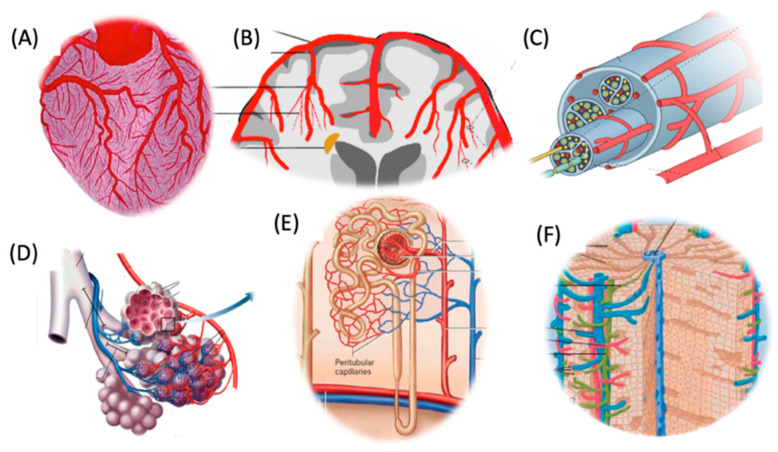
Schematic imaging of the special features of the microcirculation in major organs of the human body. (**A**) Heart; (**B**), brain; (**C**), nerves; (**D**), lung; (**E**), kidney; (**F**), liver. The illustrations are modifications of publicly available sources, including Google images.

**Figure 3 vaccines-13-00855-f003:**
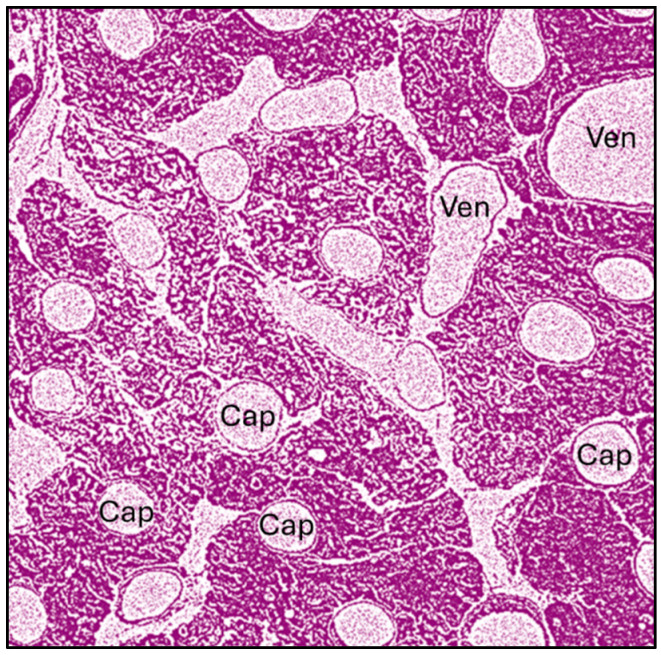
Electron micrograph of cardiac muscle fibers in a cross-section of rabbit heart. Modified from [40]. The low-magnification image (×4000) reveals the high density of microvessels. Cardiac muscle fibers are surrounded by capillaries (“Cap”) approximately 5–7 µm in diameter, with an estimated capillary-to-fiber ratio of 1:1. Post-capillary venules (Ven), measuring approximately 24–30 µm in diameter, are also visible (in the upper right corner).

**Figure 4 vaccines-13-00855-f004:**
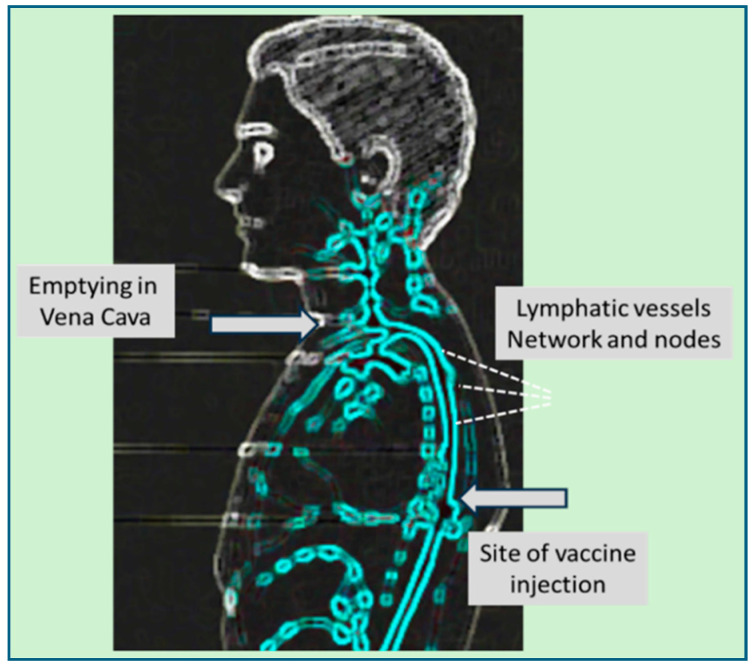
Sagittal view illustrates the cranioventral transport pathway of mRNA-LNPs from the deltoid muscle to the bloodstream via the lymphatic system. Following intramuscular injection into the deltoid muscle, the mRNA-LNPs enter the local interstitial space and are taken up by lymphatic capillaries. These merge into a network of superficial and deep lymphatic vessels whose rhythmic contractions propel the nanoparticles cranioventrally, toward the deltopectoral, lateral, central, and apical axillary lymph nodes. On the left side, this trunk drains into the thoracic duct, which empties at the left venous angle (angle of Pirogoff) at the junction of the left subclavian and internal jugular veins); on the right, drainage occurs via the right lymphatic duct into the right venous angle. (Annotated version of a standard human lymphatic system diagram from internet source).

**Figure 5 vaccines-13-00855-f005:**
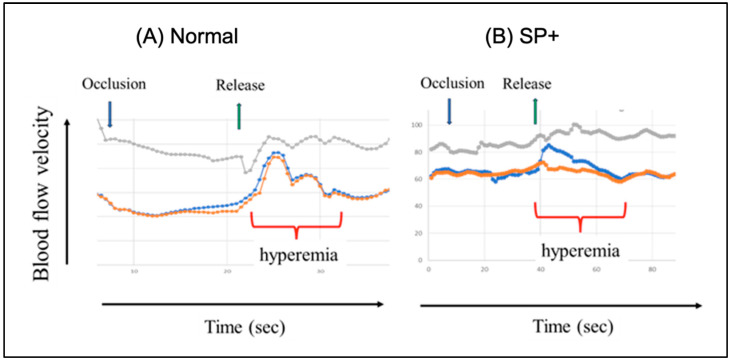
Changes in blood flow velocity in the middle cerebral arteries following 10 s manual carotid artery compression in a healthy, uninfected (**A**), and a SP+, individual showing post-COVID-19 symptoms (**B**). Blue arrow: occlusion. Green arrow: release. Red: reactive hyperemia. Gray: blood pressure. Unpublished data, collaboration with R. Debreczeni, Semmelweis University.

**Figure 6 vaccines-13-00855-f006:**
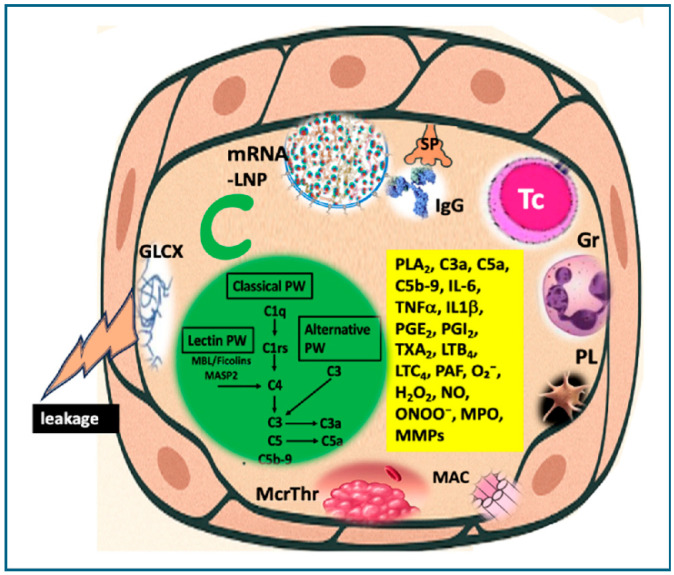
Potential processes and mediators involved in endothelitis caused by the binding and transfection of ECs by mRNA-LNPs. Activation stimuli include direct mRNA-LNP binding to ECs, complement (C) activation via anti-PEG and anti-SP IgG, and additional C activation by SP expressed on the EC membrane. Further illustrated pathological mechanisms involve autoimmune cytotoxic T cell attack; adhesion of leukocytes (neutrophils, monocytes, macrophages) and platelets to ECs injured by, among other membrane attack complex (MAC); microthrombus formation; glycocalyx degradation; and capillary leakage. The yellow box lists key inflammatory mediators contributing to these effects. Abbreviations: SP, spike protein; Tc, cytotoxic T cell; Gr, granulocyre (other leukocytes not shown); MAC, membrane attack complex; McrThr, microthrombus; GLCX, glycocalyx; C, complement cascade; O_2_^−^), superoxide; H_2_O_2_, hydrogen peroxide; NO, nitric oxide; ONOO^−^, peroxynitrite.

**Figure 7 vaccines-13-00855-f007:**
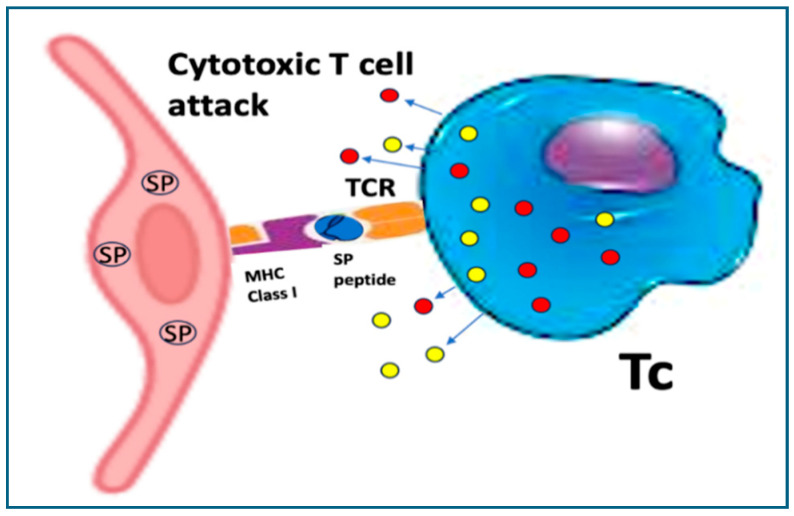
Potential autoimmune attack against ECs by spike protein-specific cytotoxic T cells with increasing impact after booster vaccinations. The scheme applies to all body cells transfected with the vaccine and expressing spike protein fragments on their surface Class-I molecules. The colored granules are cytolytic mediators, e.g., perforins and granzymes. Original drawing from a preprint [29].

**Figure 8 vaccines-13-00855-f008:**
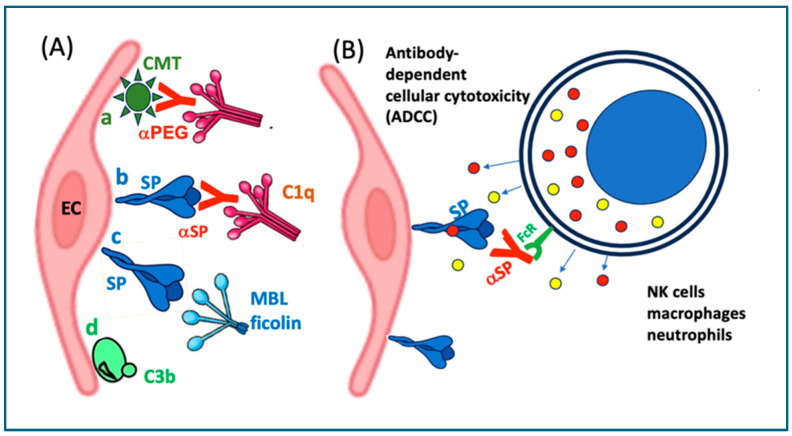
(**A**) Potential pathways of complement activation on the surfaces of endothelial cells. (**a**) The polyethylene glycol (PEG) on Comirnaty (CMT) binds anti-PEG antibodies (αPEGs) which initiate C activation via the classical pathway. (**b**) The SP expressed on the cell surface binds anti-SP antibodies and initiate C activation also via the classical pathway. (**c**) The SP expressed on the cell surface binds mannose binding lectin (MBL) or ficolin, and triggers lectin pathway C activation. (**d**) Direct deposition of C3 on membrane surfaces represents standard alternative pathway activation. (**B**) Potential antibody-dependent autoimmune attacks against endothelial cells (ECs) via antibody-dependent cellular cytotoxicity. Comirnaty; αPEG, anti-PEG antibodies; SP, spike protein; αSP, anti-SP antibody; FcR, Fcγ-receptor; MBL, mannose binding lectin. The colored granules are cytolytic mediators, e.g., perforins and granzymes [29].

**Table 1 vaccines-13-00855-t001:** Inflammatory complications associated with immunization with Comirnaty within 3 months after vaccination.

Acoustic neuritis	Herpes esophagitis	Neuropathy peripheral
Acute disseminated encephalomyelitis	Herpes pharyngitis	Neutropenic colitis
Acute encephalitis with refractory, repetitive partial seizures	Herpes simplex cervicitis	Nodular vasculitis
Acute flaccid myelitis	Herpes simplex colitis	Noninfectious myelitis
Acute hemorrhagic leukoencephalitis	Herpes simplex encephalitis	Noninfective encephalitis
Administration site vasculitis	Herpes simplex gastritis	Noninfective encephalomyelitis
Alloimmune hepatitis	Herpes simplex hepatitis	Noninfective oophoritis
Ankylosing spondylitis	Herpes simplex meningitis	Ocular vasculitis
Anti-neutrophil cytoplasmic antibody positive vasculitis	Herpes simplex meningoencephalitis	Oculofacial paralysis
Aortitis	Herpes simplex meningomyelitis	Optic neuritis
Application site vasculitis	Herpes simplex necrotizing retinopathy	Optic perineuritis
Arteritis	Herpes simplex esophagitis	POEMS syndrome
Arthritis	Herpes simplex otitis externa	Palisaded neutrophilic granulomatous dermatitis
Arthritis enteropathic	Herpes simplex pharyngitis	Pancreatitis
Atrophic thyroiditis	Herpes simplex pneumonia	Panencephalitis
Autoimmune arthritis	Herpes simplex virus conjunctivitis neonatal	Papillophlebitis
Autoimmune cholangitis	Herpes zoster meningitis	Parainfluenzae viral laryngo-tracheobronchitis
Autoimmune colitis	Herpes zoster meningoencephalitis	Paraneoplastic dermatomyositis
Autoimmune demyelinating disease	Herpes zoster meningomyelitis	Pericarditis
Autoimmune dermatitis	Herpes zoster meningoradiculitis	Pericarditis lupus
Autoimmune hepatitis	Herpes zoster necrotizing retinopathy	Peritoneal fluid protein increased
Autoimmune myocarditis	Herpes zoster pharyngitis	Peritonitis lupus
Autoimmune myositis	Human herpesvirus 6 encephalitis	Pneumonia viral
Autoimmune nephritis	Human herpesvirus 6 infection	Polyarteritis nodosa
Autoimmune pancreatitis	Hypersensitivity vasculitis	Polyarthritis
Autoimmune pericarditis	Immune-mediated adverse reaction	Polychondritis
Autoimmune thyroiditis	Immune-mediated cholangitis	Polymyositis
Autoimmune uveitis	Immune-mediated encephalitis	Primary biliary cholangitis
Autoinflammation with infantile entero-colitis	Immune-mediated enterocolitis	Proctitis herpes
Axial spondyloarthritis	Immune-mediated gastritis	Proctitis ulcerative
Bickerstaff’s encephalitis	Immune-mediated hepatic disorder	Product availability issue
Bronchitis	Immune-mediated hepatitis	Product distribution issue
Bronchitis mycoplasmal	Immune-mediated myocarditis	Pseudovasculitis
Bronchitis viral	Immune-mediated myositis	Pulmonary thrombosis
Capillaritis	Immune-mediated nephritis	Pulmonary tumor thrombotic microangiopathy
Catheter site vasculitis	Immune-mediated pancreatitis	Pulmonary vasculitis
Central nervous system vasculitis	Immune-mediated pneumonitis	Pyostomatitis vegetans radiculitis
Cerebral arteritis	Immune-mediated thyroiditis	Rash pruritic
Cholangitis sclerosing	Infected vasculitis	Rasmussen encephalitis
Chronic autoimmune glomerulonephritis	Infusion site vasculitis/thrombosis	Renal arteritis
Chronic fatigue syndrome	Interstitial granulomatous dermatitis	Renal vasculitis
Chronic gastritis	Juvenile idiopathic arthritis	Respiratory syncytial virus bronchiolitis
Chronic recurrent multifocal osteomyelitis	Juvenile polymyositis	Respiratory syncytial virus bronchitis
Colitis	Juvenile psoriatic arthritis	Retinal artery embolism
Colitis erosive	Juvenile spondyloarthritis	Retinal vasculitis
Colitis herpes	Laryngeal rheumatoid arthritis	Rheumatoid arthritis
Colitis microscopic	Leukoencephalomyelitis	Rheumatoid scleritis
Colitis ulcerative	Lichen planus	Rheumatoid vasculitis
Cutaneous vasculitis	Lichen sclerosus	SLE arthritis
Cystitis interstitial	Limbic encephalitis	Schizencephaly Scleritis
Dermatitis	Lupus cystitis	Schonlein Henoch purpura nephritis
Dermatitis bullous	Lupus encephalitis	Segmented hyalinizing vasculitis
Dermatitis herpetiformis	Lupus endocarditis	Silent thyroiditis
Dermatomyositis	Lupus enteritis	Spondylitis
Diffuse vasculitis	Lupus hepatitis	Stoma site thrombosis
Encephalitis	Lupus myocarditis	Stoma site vasculitis
Encephalitis allergic	Lupus myositis	Subacute endocarditis
Encephalitis autoimmune	Lupus nephritis	Takayasu’s arteritis
Encephalitis brain stem	Lupus pancreatitis	Terminal ileitis
Encephalitis hemorrhagic	Lupus pneumonitis	Thromboangiitis obliterans
Encephalitis periaxialis diffusa	Lupus vasculitis	Thrombophlebitis
Encephalitis post immunization	Lymphocytic hypophysitis	Thrombophlebitis migrans
Encephalomyelitis	Mastocytic enterocolitis	Thrombophlebitis septic
Enteritis	Medicaldevice site vasculitis	Thrombophlebitis superficial
Enteritis leukopenia	Meningitis	Thyroiditis
Enterobacter pneumonia	Meningitis aseptic	Tracheobronchitis
Enterocolitis	Meningitis herpes	Tracheobronchitis mycoplasmal
Enteropathic spondylitis	Meningoencephalitis	Tracheobronchitis viral
Eosinophilic granulomatosis with poly-angiitis	Meningomyelitis herpes	Tubulointerstitial nephritis and uveitis syndrome
Eosinophilic esophagitis	Mesangioproliferative glomerulonephritis	Ulcerative keratitis
Fibrillary glomerulonephritis	Microscopic polyangiitis	Urticarial vasculitis
Gastritis herpes	Mononeuritis	Uveitis
Giant cell arteritis	Myasthenic Myelitis	Vaccination site vasculitis
Glomerulonephritis	Myelitis transverse	Varicella keratitis
Glomerulonephritis membranoproliferative	Myocarditis	Varicella zoster gastritis
Glomerulonephritis membranous	Myocarditis post infection	Varicella zoster esophagitis
Glomerulonephritis rapidly progressive	Myositis	Vasculitis
Granulomatosis with polyangiitis	Nephritis	Vasculitis gastrointestinal
Granulomatous dermatitis	Neuritis	Vasculitis necrotizing
Hemorrhagic vasculitis	Neuritis cranial	meningoencephalitis
Hepatitis	Neuromyelitis optica pseudo relapse	
Herpes dermatitis	Neuromyelitis optica spectrum disorder	

**Table 2 vaccines-13-00855-t002:** Virus strains whose reactivations may be associated with inflammatory AEs caused by mRNA vaccines.

Strains	References
Cytomegalovirus (CMV)	[21,22]
Epstein–Barr virus (EBV)	[23]
Herpes simplex virus (HSV)	[24,25,26]
Varicella Zoster virus (VZV)	[27,28]

**Table 3 vaccines-13-00855-t003:** Vasculitis types and symptoms classified according to the size of the blood vessels affected.

Vessel Size	Types of Vasculitis	Typical Symptoms
Small	Microscopic polyangiitis (MPA) Granulomatosis with polyangiitis (Wegener’s) Eosinophilic granulomatosis with polyangiitis (Churg-Strauss) IgA vasculitis (Henoch–Schönlein purpura) Cryoglobulinemic vasculitis Hypocomplementemic urticarial vasculitis (anti-C1q vasculitis) Cutaneous leukocytoclastic vasculitis Leukocytoclastic vasculitis ANCA *-associated vasculitis	Vasodilation Increased vascular permeability Leukocyte adhesion, margination and transmigration Endothelial activation and dysfunction Microthrombus formation Pain tissue hypoxia/ischemia lactate accumulation, acidosis
Medium	Polyarteritis nodosa (Kawasaki disease) ANCA-associated vasculitis	Skin ulcers Abdominal pain Mononeuritis multiplex Hypertension
Large	Giant cell arteritis Takayasu arteritis	Headache Visual disturbances Jaw claudication Limb claudication Elevated ESR/CRP

* ANCA, anti-neutrophil cytoplasmic antibodies; ESR, erythrocyte sedimentation rate; CRP, C-reactive protein. Information collected, among others, from Refs. [83,84,85,86].

**Table 4 vaccines-13-00855-t004:** Selected pathological changes observed in endothelitis and their potential contributing mechanisms and consequences.

Abnormality	Contributing Mechanisms	Potential Consequences	References
Endothelial cell activation	Complement and recruited leukocyte/platelet activations Release of anaphylatoxins, proinflammatory cytokine, lipid and enzyme mediators, danger-associated molecular patterns, ROS liberation, exosome secretion	New-onset or flares of vasculitis -> acute and chronic multiorgan inflammatory reactions Edema Skin alterations Multi-organ dysfunction Triggering of ARDS Autoimmune phenomena	[109,110,111,112,113,114,115,116,117,118,119,120,121,122,123,124,125,126,127,128,129,130,131,132,133]
Vascular injury	Glycocalyx degradation -> EC membrane injury Disruption of tight junctions -> increased vascular permeability Capillary occlusion -> perfusion blockage or hypoperfusion Reduced nitric oxide (NO) production -> hypertension		
Thrombogenesis	Exposure of collagen and tissue factor -> microthrombus formation Upregulation of adhesion molecules (e.g., ICAM-1, VCAM-1) -> platelet activation/aggregation Downregulation of thrombomodulin and NO -> vasoconstriction -> Platelet activation/aggregation Coagulation Cascade Activation Thrombus formation Leukocyte–platelet aggregation -> microcirculation blockage -> hypoperfusion	Atherosclerosis Pulmonary embolism Stroke Myocardial infarction Disseminated intravascular coagulation (DIC) ARDS Deep vein thrombosis	[47,119,134,135,136,137,138,139,140,141]
Autoimmune damages	Expression of neoantigens on endothelial membrane Cross-presentation of SP Release of DAMPs autoreactive T and/or B cell production Persistent immune activation	New-onset or flares of autoimmune disorders Chronic autoimmune disorders	[127,128,142,143,144,145]

Abbreviations: ARDS, acute respiratory distress syndrome; DAPM, danger-associated molecular pattern; EC, endothelial cell; ICAM-1, intercellular adhesion molecule-1; VCAM-1, vascular cell adhesion molecule-1; ROS, reactive oxygen species; NO, nitric oxide; PLA2, phospholipase A2; SP, spike protein.

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
