# Peer review of "Multisystem Endothelial Inflammation: A Key Driver of Adverse Events Following mRNA-Containing COVID-19 Vaccines"

_vaccines, 2025, doi:10.3390/vaccines13080855_

Round 1

Reviewer 1 Report

Comments and Suggestions for Authors

The manuscript provides a thorough review of the potential mechanisms underlying the adverse reactions to mRNA-containing COVID-19 vaccines, particularly focusing on microcirculatory inflammation. It incorporates recent findings and data, such as the preclinical study by Pfizer Australia on the biodistribution of mRNA-LNPs. It also highlights the distinct microcirculatory architectures across different organs and their implications on vaccine-induced inflammation.

Major comments:

1# While the manuscript provides a comprehensive overview of the potential adverse reactions to mRNA vaccines, it may overemphasize the incidence and severity of these events. Although it acknowledges that AEs are statistically rare, the extensive discussion on the various inflammatory complications could potentially create a biased perception of the overall safety profile of mRNA vaccines.

2# 

Some of the proposed mechanisms and pathways, such as the role of the SP in causing endothelial dysfunction and the potential for mRNA-LNPs to trigger autoimmune responses, are based on speculative hypotheses. While these hypotheses are intriguing and warrant further investigation, they should be clearly distinguished from established facts.

3# The manuscript could benefit from a more critical evaluation of the evidence supporting these hypotheses, including a discussion of the limitations and uncertainties associated with the current understanding of mRNA vaccine-induced inflammation.

4# The authors may like to supplement a discussion on how the safety profile and adverse event spectrum of mRNA vaccines compare to those of other vaccines would provide a more comprehensive understanding of the overall landscape of COVID-19 vaccination.

5# The manuscript references several online materials, such as the 3-month safety surveillance report of Pfizer. However, it is essential to ensure that all online materials are from reliable sources and are appropriately cited and referenced to avoid any potential biases or inaccuracies.

6# It is crucial to continuously update the manuscript with the latest research findings and publications to maintain its relevance and accuracy. The manuscript aligns much with recent publications on the topic of mRNA vaccine safety and adverse events. 

7# There are publications focus on AEs on mRNA vaccine. Authors should make considerable discussion and appropriately cite those advancing studies. such as:

1)Myocarditis following immunization with mRNA COVID-19 vaccines in members of the US military JAMA Cardiol., 6 (2021), pp. 1202-1206

2)Adverse events following COVID-19 mRNA vaccines: a systematic review of cardiovascular complication, thrombosis, and thrombocytopenia Immun. Inflamm. Dis., 11 (2023), Article e807

3)[Illuminating the shadows - perspectives on mRNA vaccine adverse events - mechanisms, risks and management]: A review. Int. J. Biol. Macromol., 318 (2025), 145010

8# The extensive discussion on rare and severe adverse events could potentially be misinterpreted by readers, leading to undue concerns about the safety of mRNA vaccines. It is essential to ensure that the manuscript's content is presented in a manner that accurately reflects the current scientific consensus and does not contribute to misinformation.

9# In the methods section, clearly define the databases for literature search, keywords, and the time range of retrieval. Develop and elaborate on the criteria for including and excluding individual studies, such as study types and sample size requirements. Author may like to refer to the following examples:

1)https://doi.org/10.1016/j.ijbiomac.2025.145010;

2)Post-vaccination Syndrome: A Descriptive Analysis of Reported Symptoms and Patient Experiences After Covid-19 Immunization, medRxiv

10# The discussion on the causes of LNP complications is too brief and lacks thorough elaboration. Additionally, other factors influencing microcirculatory inflammation—such as viral nucleic acids stimulating the body's inflammatory response—were not fully discussed.

Minors issues,

"mRNA vaccine" is ok. No need to use "mRNA-based/associated/related vaccine".

Is it necessary to use both PVS and AEs. If so, must to tell the differences between them and discuss each piece separately; if not, keep only one of them.

Reviewer 2 Report

Comments and Suggestions for Authors

I sincerely appreciate and I would like to thank the Authors for the high-quality summary of this important topic, as COVID-19 itself is a life-threatening clinical condition and vaccintation against it should be clarified and improved as much, as possible also thanks to the clarification of its adverse effects.

Content suggestions:

  1. For the completness, can the Authors specify the epidemiology of PVS ?   
  2. Can the Authors outline the most important feature of PVS for

the patients – its management ?

Reviewer 3 Report

Comments and Suggestions for Authors

This manuscript brings a review about the background of adverses reactions of mRNA COVID-19 vaccines.

The topic is of interest as can help to prevent and improve the outcome in these adverse reactions.

The review is very detailed and focusses in the diverse mechanisms leading to microcirculatory inflammation. Some of the paragraphs are a little dense even though I understand that the mechanism is complex and need a detailed explanation.

Some of the figures do not include their origin.

Table 3- The bullets and text is not aligned.

Reviewer 4 Report

Comments and Suggestions for Authors

Congratulations! - I was mesmerized with this paper - almost written like a thriller. I think it is a very important article - reads like a text book description. The figures were spot on, very clear and very well done. 

I got a bit lost along the way with the number of acronyms - perhaps you could repeat some full words in between - otherwise one is going back and forth for some of them. 

Also, I felt section 10 should have been right up in front. 

Is it possible to have some sub sections - otherwise there is so much going - for example in Section 9 - you start off with Spike protein and you end up with something else - if there were sub headings or sub sections - it might help us read it better. 

I think this belongs in a text book or some learning material for all PGs etc. 

Reviewer 5 Report

Comments and Suggestions for Authors

Comments and Suggestions

Section Title:
1. It should be indicated that this is a narrative review to avoid confusion with systematic reviews or original studies.
Introduction Section
2. Lines 40-50 are repeated almost verbatim (double paragraph on the review of CDC guidelines and the incidence of adverse events).
3. Eliminate duplication and condense the justification into a single paragraph.
4. The manuscript assumes the existence of "Post-Vaccination Syndrome (PVS)" as a premise, a term not yet agreed upon and supported by preprints and gray literature (refs. 1-3).
5. Define PVS operationally, differentiate association from causality, and add peer-reviewed systematic reviews or meta-analyses that support the concept.
6. It is stated that "officially reported" AEs range from 0.03-0.5% without citing the primary source or clarifying the denominator (serious vs. total AEs). Add a primary reference (e.g., VAERS, EudraVigilance) and compare with population-based baseline incidences.
Methodology Section
7. The manuscript lacks a methodology section indicating the selection criteria for studies for the review and the type of review.
Main Section
8. Table 1: The list of inflamed organs lacks validation and includes heterogeneous entities.
9. Describe inclusion criteria, add manual review, and reduce AEs to those with robust epidemiological evidence.
10. Extrapolate animal biodistribution data (rats, pigs) to suggest systemic endothelial inflammation in humans.
11. Introduce caution when extrapolating with human biodistribution (e.g., PET-CT or autopsies) and mention limitations.
12. Risk-Benefit Balance: The manuscript overemphasizes toxicity without contextualizing or contrasting the reduction in mortality and hospital burden attributed to vaccines.
13. Include a paragraph summarizing effectiveness data and placing the AEs in the current population context.
References
14. More than 20% are preprints or opinion articles; some are not indexed publications.
15. This is because the search criteria or inclusion of references have not been defined.
16. Replace with peer-reviewed studies.
17. Verify Vancouver format.
General review:
18. Long sentences, inconsistent use of tense, and redundant technical vocabulary.
19. Break up sentences, simplify syntax, and standardize terms (e.g., “lipid nanoparticles” vs. “LNP”).
Figures and tables
20. Figures should have explanatory footnotes.
21. Table 3 reproduces very long text within the body.
22. Move long tables to supplementary material.

Round 2

Reviewer 1 Report

Comments and Suggestions for Authors

The comments are mostly well-addressed. However, for 9# and minor point of "PVS and AEs", authors may not fully understand the referee's point. 

9# means the methods should be on the basis of solid evidence, even that it is a review. Since chatGPT is evolving and developing to different version, appropriate strategy would be based on official academic database. It does not mean that chatGPT is not allowed but better only for reference as a mirror instead. This is for the sake of scientific rigor.

As for "PVS and AEs", it is a similar situation like "mRNA-based/associated/related vaccine" from the referee's view. If author intended to use PVS and AEs for different object, it would be ok to keep both of them.

Author Response

Reviewer: The comments are mostly well-addressed. However, for 9# and minor point of "PVS and AEs", authors may not fully understand the referee's point.

9# means the methods should be on the basis of solid evidence, even that it is a review. Since chatGPT is evolving and developing to different version, appropriate strategy would be based on official academic database. It does not mean that chatGPT is not allowed but better only for reference as a mirror instead. This is for the sake of scientific rigor.

Response: We thank the Reviewer for the valuable suggestion to include a methodological section detailing our search terms, data sources, and publication date range, similar to the format used by Kang et al. in their review “Illuminating the Shadows – Perspectives on mRNA Vaccine Adverse Events” (DOI: 10.1016/j.ijbiomac.2025.145010). However, due to the exceptionally diverse set of search terms and data sources in our review, as well as the broad range of publication dates, it was not feasible to present this information in the same short yet exhaustive, structured manner. Instead, we revised the sentence in line 68 to clarify our approach: “This review, based on official academic databases, focuses on microcirculatory inflammation…”

Reviewer: As for "PVS and AEs", it is a similar situation like "mRNA-based/associated/related vaccine" from the referee's view. If author intended to use PS and As for different object, it would be ok to keep.

Answer: Yes, we wish to distinguish the two terms, and use them in the text as appropriate. We thank the Reviewer for agreeing to this.